# Mass-Accreting Pulsating Components of Algols

David Mkrtichian [1,*,†], Khemsinan Gunsriviwat [1,2,3,†], Holger Lehmann [4], Chris Engelbrecht [5],
Andrew Tkachenko [3] and Victor Nazarenko [6]

1   National Astronomical Research Institute of Thailand, 260 Moo 4, T. Donkaew, A. Maerim,
    Chiang Mai 50180, Thailand
2   Department of Physics and Materials, Faculty of Science, Chiang Mai University, Chiang Mai 50200, Thailand
3   Institute of Astronomy, KU Leuven, Celestijnenlaan 200D, B-3001 Leuven, Belgium
4   Thüringer Landesternwarte Tautenburg, Sternwarte 5, 07778 Tautenburg, Germany
5   Department of Physics, University of Johannesburg, Auckland Park, P.O. Box 524,
    Johannesburg 2006, South Africa
6   Astronomical Observatory, Odessa National University, 650014 Odessa, Ukraine
*   Correspondence: davidmkrt@gmail.com; Tel.: +66-806716680
†   These authors contributed equally to this work.

**Abstract:** We present a review of the latest results of studies of the class of mass-accreting pulsating components of semi-detached eclipsing binaries known as oEA stars. The application of the techniques of asteroseismology to this class of stars unlocks new pathways for gaining a deeper understanding of the short-term evolution and magnetic activity of binary stars. We report the discovery of 49 new pulsating components of eclipsing binaries, based on data from NASA's TESS space telescope. Recent observational results on the pulsation characteristics of these stars are summarized. The effects of the interaction of the magnetic and spot activity of the Roche-lobe-filling component of a system with the pulsations of the mass-accreting component are discussed.

**Keywords:** stars; binaries; eclipsing-stars; delta scuti stars; oscillatingstars

## 1. Introduction

Various classes of pulsating A- and F-type stars co-exist in the lower domain of the classical instability strip on the HR diagram (e.g., $\delta$ Sct, $\gamma$ Dor, etc.). The pulsations in these stars are driven mainly by the kappa-mechanism in the partial ionization zone associated with the second ionization of helium in the stellar envelope. Many pulsating Am stars, $\lambda$ Boo stars, $\gamma$ Doradus stars, pulsating pre-main sequence stars, magnetic Ap stars, and pulsating mass-accreting components of semi-detached Algol systems, with a variety of pulsation characteristics, have been discovered in recent decades, augmenting also the well-known class of classical $\delta$-Scuti-type stars [1] in the hotter end of the classical instability strip.

Stellar surveys have shown that approximately two-thirds of the stars in our Galaxy are members of binary or multiple systems [2]. Therefore, the co-existence of stellar pulsations and binarity or multiplicity among A-F stars is expected to be a common occurrence. A recent systematic survey of pulsating components among approximately 3000 *Kepler* eclipsing binaries (EBs) [3] has led to the detection of 303 systems (about one-tenth of the sample) whose light curves display evidence of oscillations. Among these 303 systems, 149 stars were flagged as $\delta$ Scuti stars, 115 as $\gamma$ Doradus stars, 59 as tidally excited pulsators, and 85 as evolved red giant stars.

EBs are observationally selected binary systems with orbital planes that fall within a few degrees of containing the line of sight. The spectroscopic orbits of double-lined EBs and their photometrically detected periodic eclipses permit the derivation of accurate physical dimensions of the individual components and the orbital elements, as well as model-independent dynamical masses, with a precision of approximately one percent [4]—one

order of magnitude better than that obtained for single stars. The accurately determined physical parameters of binary components can be used as direct constraints in the search for relevant models of stellar structure and evolution across the HR diagram. Furthermore, when pulsations of either one or both components of eclipsing binary systems are detected, the accurately determined physical parameters can be used very effectively for the refinement of the pulsation model.

The Algol-type systems (Algols, hereafter) are semi-detached interacting eclipsing binary systems consisting of a mass-accreting main sequence star of the B, A, or F (the primary, hereafter) spectral type and an evolved F-K-type subgiant component (the secondary, hereafter). The first pulsating $\delta$-Scuti-type components identified in Algols, AB Cas [5], and Y Cam [6] were discovered in the 1970s. However, it was only established in the early 2000s that they belonged to a new class of pulsating stars. The existence of a previously unrecognized new class of A- or F-type pulsating, mass-accreting components of semi-detached eclipsing binary systems was claimed by [7]. In 2004, for classification purposes, [8] proposed that this class should be called the oscillating Eclipsing Algol (oEA) stars in order to differentiate them and confirm that they are, in evolutionary terms, different from both the single classical $\delta$ Scuti stars and the $\delta$ Scuti pulsators in well-detached binary systems designated in the General Catalogue of Variable Stars (http://www.sai.msu.su/gcvs/gcvs/ (accessed on 04 July 2022)) as EA/DSCT. A particular oEA star with the pulsating component is depicted in Figure 1.

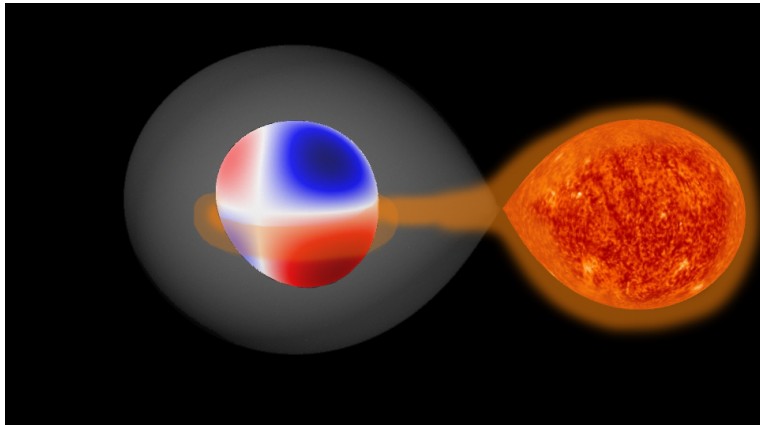

**Figure 1.** A simulation of the Algol system RZ Cas in accordance with its binary parameters and the primary component's nonradial pulsations. The primary, which is also the mass-accreting component, is inside its Roche lobe, while the secondary fills its Roche lobe. A superposition of three pulsation modes with l,m = 2,1, l,m = 3,2, and l,m = 1,0 at the surface of the primary is shown.

We present a concise review of the most recent observational data and analytical results associated with pulsations identified in eclipsing binaries observed by TESS and in a representative well-studied case (RZ Cas). We also discuss new observational evidence pointing to the influence of magnetic activity cycles on the mass transfer rate and pulsational properties of mass-accreting components. We further review new spectroscopic and photometric indicators of seasonal variations in mass transfer, discuss the possible mechanism controlling the variations of the rate of mass transfer, and demonstrate how changes in the pulsation spectrum correspond to seasonal variations of the mass transfer rate.

## 2. The Evolutionary Status of oEA Stars and Their Importance for Studies of Short-Term Binary Evolution

The mass-accreting primary components of semi-detached Algol systems (with spectral classes A–F) passed through their first rapid stage of mass accretion during their evolution from a detached to a semi-detached binary configuration. During the rapid mass transfer stage, the (initially) low-mass components were "re-fueled" by an additional and essential amount of hydrogen transferred from the outer volume of the (initially more

massive) mass-losing components, which, during their own evolution, started to exceed the boundaries of their Roche lobes. During the stage of rapid mass transfer, the (initially) low-mass components rapidly (that is, on the thermal timescale) re-adjusted their internal structure while eventually becoming more massive than their donor companions and moving towards more luminous and hotter domains on the HR diagram, specifically, into the domain defining the lower (hotter) end of the classical instability strip on the main sequence (MS). The details of angular momentum transfer driving the evolutionary pathways of Algols during this stage are still not completely understood. A comprehensive overview of stellar conditions during this rapid, first mass transfer stage can be found in [9], wherein mass loss rates associated with different proposed mechanisms of systemic mass loss are compared. That study revealed that the "hotspot" model [10–12], in which mass loss is driven by the radiation pressure of a hot spot, was the most realistic model and that mass transfer during this stage is non-conservative, with an amount of mass from the donor star escaping entirely from the system.

Following the first, rapid mass transfer stage, a slow mass transfer stage ensues. If the primary component has settled (following the initial mass transfer episode) into a new location in the HR diagram that lies inside the classical instability strip, it can undergo $\delta$-Scuti-type pulsations similar to those of the classical single A- to F-type pulsators and of pulsators in detached systems, which occupy the same part of the classical instability strip. The pulsational properties of single short-period A-F pulsators are well described in recent reviews (see, e.g., [1]). The oEA pulsators, on average, share almost all their physical characteristics (mass, radius, luminosity, chemical abundances, etc.) with the single A-F main sequence pulsators. The difference between oEA stars and other single A-F main sequence pulsators lies in their prior evolutionary pathways and in the ongoing mass accretion that keeps them in thermal imbalance and directs them onto evolutionary pathways that differ from those of single A-F stars.

The observational data collected during a century of spectroscopic and photometric investigations of Algols confirm that long-term secular changes in orbital periods result from the mass transfer episodes, whereas the alternate decades-long (6 to 40 years) cyclic variations in orbital periods can potentially be explained by magnetic activity cycles, as proposed by [13], or by the existence of the tertiary component. Applegate's hypothesis suggests that the magnetic activity cycles in magnetically active components of Algols and other types of binary systems modify the geometries of the active components and also re-distribute angular momentum inside those components. These processes result in changes in orbital angular momentum, the distance between the components, and the orbital period of a system. In an Algol system, the resulting changes in the distance between the components should also influence the mass transfer rate from the secondary component (the donor, hereafter) to the primary component (the gainer, hereafter), which, in turn, affects the orbital period in a cyclical process. The role of the relocation of the L1 point in the atmosphere of the donor in affecting mass transfer resulting from magnetic activity has not been studied in depth as of yet. This is because of the difficulty in estimating the actual moment-to-moment mass transfer rate from observational data, the accurate determination of the position of the L1 point, and many other unknown parameters influencing the mass transfer rate. Realistic numerical simulations of mass transfer rate variations using model atmospheres for the donor and hydrodynamic codes based on the "large-particle" numerical method [14] were performed by [15]. In the latter study, the authors used Kurucz's [16] model atmospheres for a K-type donor and moved the atmosphere outward and inward with respect to the L1 point to vary the mass transfer rate. Their calculations demonstrated that the degree of Roche lobe overflow and the rate of mass transfer observed in Algols are dependent on the height of the location of the inner Lagrangian point in the atmospheric layers of the donor.

The variations in the height of the L1 point in the modeled atmosphere of the donor led to concomitant variations in the mass transfer rate. In the direct gas stream atmosphere impact systems, the gas stream flowing from the L1 point either impacts the equatorial

zones of the pulsating star (the gainer) or accretes onto the atmosphere from an accretion disk. In either case, the gas stream transfers a substantial amount of angular momentum to the gainer and forces the acceleration of its superficial layers, causing strong differential rotation in the atmospheric layers of the gainer. Variations in the extent of differential rotation can alter the frequencies of the tesseral and sectoral pulsation modes, which can be analyzed with very high precision using asteroseismology.

In previous classifications of oEA stars [7,8], it was suggested that non-stationary mass transfer in an Algol, including accretion, modifies the physical conditions in the outer envelope of the gainer. If the gainer is pulsating, changes in its physical conditions would affect the excitation of pulsation modes, which are highly sensitive to conditions in the outer stellar envelope. Theoretical analyses are not yet available, but it is very likely that episodes of accretion would influence the spectrum of the excited modes of stellar acoustic oscillations and that the characteristics of the excited modes would be detectable through observation (further considerations accompany the discussion of the RZ Cas system in Section 5).

Therefore, in the case of the oEA pulsators, we may expect to observe the interactions between magnetic cycles in the donors and pulsations of the mass-accreting gainers and, specifically, how such interactions are associated with the occurrence of variable mass transfer and accretion. The latter can influence the surface differential rotation and the physical conditions in the atmosphere of the gainer and generate variations in excited pulsation frequencies and mode spectra that would be observable and accurately measurable using precise space-based photometry and ground-based spectroscopy.

These perspectives confirm why the group of oEA stars is of unique asteroseismic interest. Studies of these stars open a new window for monitoring the short-term evolution of interacting binary systems via the detection of variations of the acoustic spectra of oEAs.

The interrelation of a sequential chain of events: cycles of magnetic activity—changes in the shape of the Roche-lobe-filling donor—immediate variations in the orbital period and in the separation of the components—mass transfer instabilities—have not yet enjoyed intensive observational investigation. The exact mechanism triggering the mass transfer instability in the Roche-lobe-filling donor remains an open question, which has stimulated studies of Algols (especially through spectroscopy) on a case-by-case basis, notwithstanding the difficulty of detecting the observational manifestations of these events. Consequently, there is a need for detailed observations of selected magnetically active Algols and for novel observational indicators of magnetic cycles and strong mass transfer events.

To prove the effectiveness of mass transfer in influencing pulsations, the following chain of events needs to be established:

- Clearly detect evidence of magnetic activity via direct detection of magnetic fields and their variations or, implicitly, through the cyclic behavior of variations in the orbital period or through the detection of other observational phenomena.
- Obtain (through photometric, spectroscopic, or other observational techniques) clear evidence of variations in mass transfer rates and prove that such variations were caused by magnetic activity cycles.
- Clearly detect corresponding changes in pulsation spectra and prove that variations in mass transfer/accretion rates caused such changes.

The realization of this chain of events requires the well-organized spectroscopic and photometric monitoring of selected oEA stars for the duration of at least one full magnetic activity cycle. The following sections include a general description of the pulsational characteristics of oEA stars and a discussion of the specific example of a key active Algol, RZ Cas. The results of monitoring campaigns focused on RZ Cas are also presented. We demonstrate new observational indicators of variable mass transfer events and discuss the associations between the mass transfer events and the pulsational variability of the primary component.

## 3. The Pulsational Parameters of oEA Stars

Subsequent to the class of oEA stars being formally established, the number of mass-accreting components discovered from the ground showed a steady increase, driven by various survey programs (e.g., [17–21]. The majority of oEA stars are multiperiodic non-radial pulsators (NRPs) with periods from about 20 min to several hours (see Table A1 in [22]), usually with amplitudes lower than 10 mmag [23]. Following an earlier study by [21,23], an up-to-date catalog of 92 Delta-Scuti-type stars discovered in detached binary systems, as well as in semi-detached systems was presented. The pulsational properties of these two groups of stars were found to be very similar, with the oEA stars displaying slightly shorter pulsation periods than the pulsators in detached systems. This difference between the groups may be attributed to the fact that the evolutionary history of pulsating components in Algols render them closer to the main sequence, overall, than pulsators in detached systems. Reference [22] listed more than 70 identified oEA stars.

Because of the limited precision of ground-based observations, ground-based searches for pulsations in oEA stars have been biased toward the detection of high-amplitude pulsators. Observations by the space-based *Kepler* and TESS missions have improved the detection precision by two to three orders of magnitude, leading to a doubling of the number of identified oEA stars. The pulsation spectra of many oEA stars are now well resolved. Reference [24] listed 57 new pulsating components in EA-type binary systems by analyzing TESS data. In Table 1, we present a list of 49 new oEAs and some detached pulsating eclipsing binary stars discovered by us since 2018, based on TESS light curves. Figure 2 shows the TESS binary light curve of RR Dra containing very-low-amplitude oscillations hidden in the observational scatter. Figure 3 displays the discrete Fourier transform (DFT) spectrum of the out-of-eclipse light curve of RR Dra after the variations caused by the binary motion were removed. Multi-periodic very-low-amplitude (0.36 mmag) pulsations in the primary A2 V component are exhibited in this DFT spectrum.

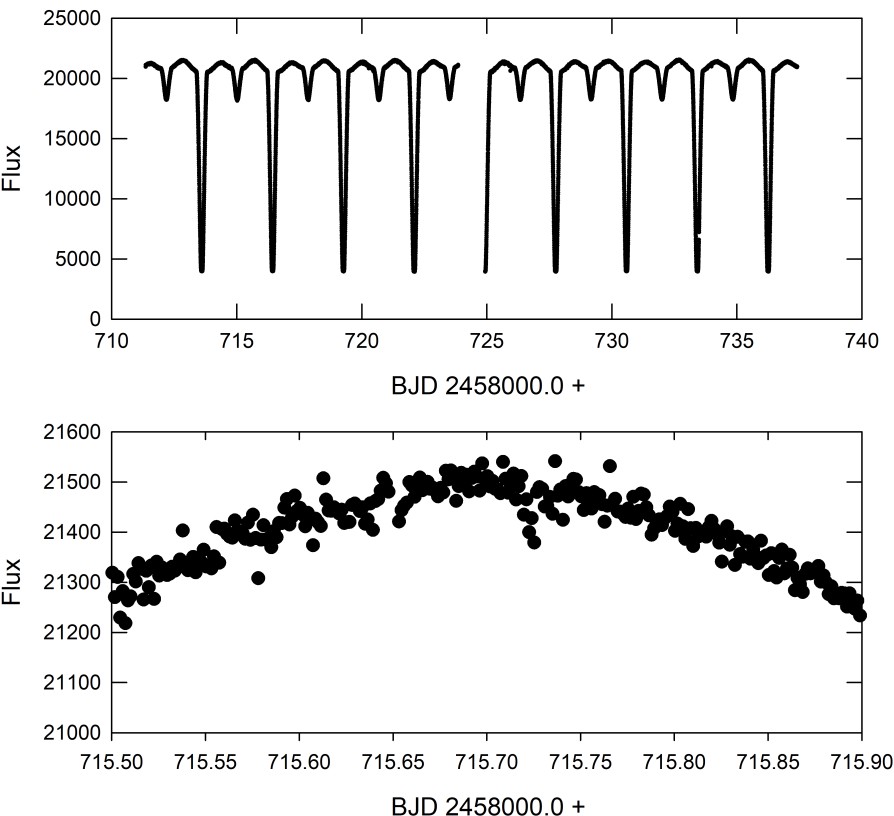

**Figure 2.** (**Top panel**) The 24-day-long TESS light curve of the new oEA star RR Dra. (**Bottom panel**) Zoomed-in portion of the light curve of RR Dra with the 36.7-minute low-amplitude oscillations of the primary component hidden in the photometric noise.

**Table 1.** The list of 49 new oEA and EA/DSCT stars discovered by us in the TESS light curves. A number of the pulsating components of eclipsing systems have uncertain detached/semi-detached configurations; they need further clarification of their binary configurations. The pulsation frequency $f_{puls}$ is given for the mode of the highest photometric amplitude. The orbital periods and pulsation frequencies are listed with five- and two-digit precision, respectively.

| System | TESS Name | $P_{orb}$ (days) | Type | Sp | $f_{puls}$ (c/d) |
|---|---|---|---|---|---|
| RZ Hor | 31653503 | 6.67999 | EB | F | 22.25 |
| VX Hyi | 33834253 | 3.23220 | EA/SD | F4 | 9.06 |
| VX Cet | 35743561 | 2.72076 | EA/SD | F7 | 21.70 |
| IQ CMa | 37601240 | 0.73138 | EB | A8V | 17.90 |
| GI Boo | 68032870 | 1.03349 | EB | A5 | 40.05 |
| SX Lyn | 81038220 | 2.02249 | EA/SD | A2 | 31.00 |
| SW Phe | 120414806 | 2.55310 | EA/SD | A5 | 39.90 |
| HD 23692 | 121078334 | 0.92833 | EB | A4IV | 17.68 |
| RY Ind | 126602778 | 0.71211 | EB | A5 | 17.27 |
| SU For | 129764561 | 2.43461 | EA/SD | A2 | 25.24 |
| CH Ind | 139699256 | 5.9532 | EA | A9V | 8.85 |
| RX Pic | 150443185 | 2.59365 | EB | A2/3V | 45.98 |
| GK Eri | 156215585 | 2.95966 | EA | F0 | 9.13 |
| GL Boo | 158016784 | 3.19738 | EA | - | 7.82 |
| TY UMi | 159298033 | 1.72488 | EA | F0 | 18.04 |
| V548 Cyg | 165310952 | 1.80523 | EA/SD | A1V+F7 | 7.59 |
| CD-31 1621 | 166874908 | 2.18675 | EB | A3 | 28.88 |
| RY Gru | 175405906 | 2.01063 | EA/SD | - | 18.22 |
| V392 And | 176854066 | 4.04628 | EA | A2 | 9.53 |
| HD29766 | 178996712 | 2.98240 | EA | A2mA8-F3 | 28.77 |
| V Tuc | 181043970 | 0.87092 | EA/SD | A2IV | 58.79 |
| HD26306 | 198037741 | 0.78003 | EA | A4V | 20.45 |
| HD30204 | 200440270 | 1.07871 | EA | A2IV/V | 60.66 |
| X Pic | 219373406 | 0.86190 | EA/SD | A2 | 50.33 |
| AN Tuc | 231714000 | 5.46132 | EA/SD | A5III | 32.42 |
| HD 160862 | 233195058 | 2.67932 | EA | A2 | 20.53 |
| RR Dra | 233532554 | 2.83128 | EA/SD | A2 | 39.23 |
| EQ Ori | 244250449 | 1.74605 | EA/SD | A0 | 38.22 |
| Y Hyi | 262958558 | 3.53597 | EA/S | A6V | 47.30 |
| GH Cet | 266735682 | 1.13524 | EA | A5 | 51.14 |
| AI Dra | 274509791 | 1.19882 | EA/SD | A0V | 18.35 |
| HD 54011 | 279569707 | 3.97936 | EA | A1/2A5 | 12.92 |
| CD-70 152 | 280831485 | 14.14 | EA | F1 | 22.79 |
| AK Dra | 289722957 | 2.21830 | EA | F3 | 22.78 |
| W Vol | 300654002 | 2.75836 | EA | F1V | 19.39 |
| TX Vol | 310308203 | 5.38837 | EA/D | A3 | 13.99 |
| XZ UMa | 318217844 | 1.22230 | EA/SD | A5+F9 | 48.54 |
| WZ Pic | 350443417 | 1.21669 | EB | A2mA7-A9 | 22.78 |
| V706 And | 352077081 | 2.52094 | EA | A | 30.70 |
| HD 33717 | 358335586 | 1.22041 | EA | A0V | 37.90 |
| HD 43898 | 393387739 | 3.07031 | EA/D | A8/9V | 21.32 |
| V707 And | 396134795 | 2.58863 | EA | F5V | 14.20 |
| ES Ori | 397048159 | 1.60556 | EA/SD | A2V | 15.73 |
| AX Vul | 406421379 | 2.02484 | EA | A1V | 75.55 |
| V629 And | 428003183 | 0.7426 | EA | A | 22.70 |
| V343 Lac | 430808126 | 6.47191 | EA/SD | A0V | 17.01 |
| NN Cep | 434625997 | 2.05830 | EA | A5 | 8.28 |
| TYC 683-640-1 | 450089997 | 2.46305 | EA | F | 13.87 |
| UY Vir | 452734608 | 1.99445 | EA/D | A7V | 17.17 |

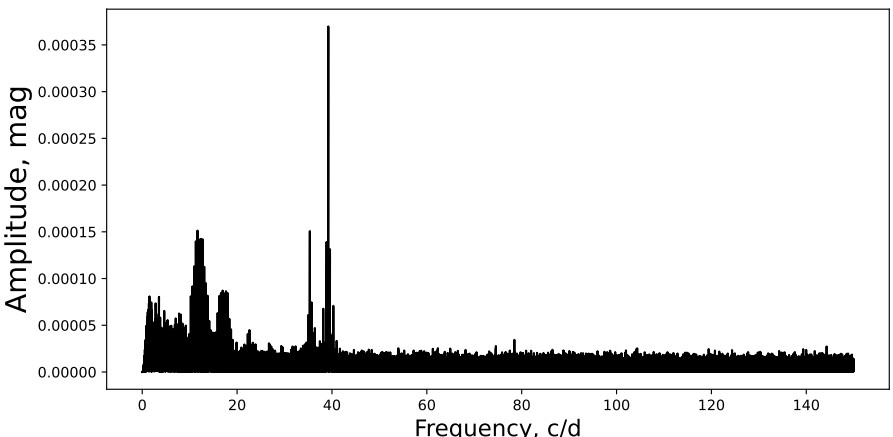

**Figure 3.** The discrete Fourier transform spectrum of the out-of-eclipse light curve of RR Dra, displaying multiple frequency peaks. The peak at 39.22 c/d (corresponding to a period of 36.7 min) with an amplitude of 0.000369 mag is associated with the dominant oscillations of the primary component.

## 4. Low- and High-Degree Nonradial Pulsations

The oEA stars have been discovered in observationally selected eclipsing binary systems. As components of eclipsing systems, we expect that these stars are being viewed close to equator-on. This is the optimal inclination to the line of sight for the detection of sectoral (l = |m|) NRPs when the pulsational axis of symmetry is aligned with the rotation axis, as these modes reach their maximum amplitudes around the stellar equator. Strong confirmation of this expectation has now been established for a sample of Algol stars observed spectroscopically. Reference [25] published a preliminary report on a high-resolution spectroscopic survey of 24 oEA stars made with the SALT and the 2.4 m Thai National Telescope, presenting detections of high-degree nonradial pulsations in line profile variations of a sample of 7 oEA stars.

Figure 4 shows the observed spectral line profile variations of the oEA star AS Eri calculated using the least-squares deconvolution (LSD) code designed for binary stars by [26], which correspond to prograde tesseral pulsation modes of a high degree. The spectroscopic detection of high-degree modes in addition to modes detected through photometry (which is sensitive to low-degree modes) indicate that a broad spectrum of low- and high-degree nonradial pulsation modes is excited, offering attractive prospects for the intensive asteroseismology of oEA stars.

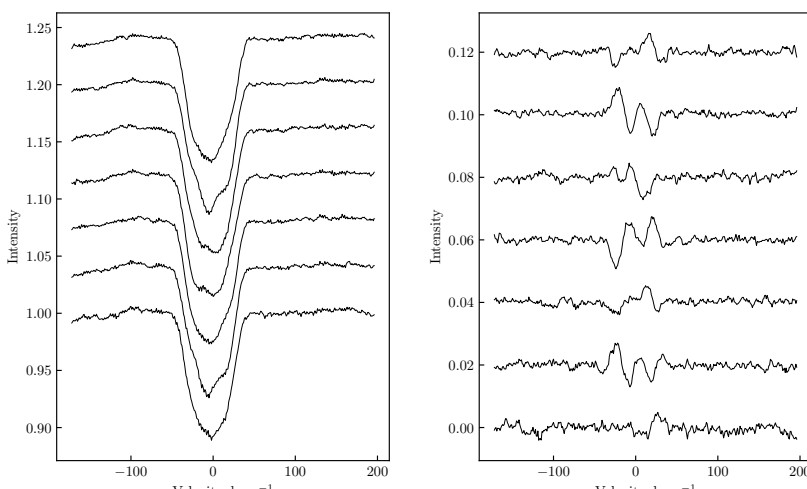

**Figure 4.** The pulsational least-squares deconvolution (LSD) line profile variations in AS Eri observed with the SALT telescope. (**Left panel**) Pulsational variations of LSD profiles; (**right panel**) the residual line profile variations.

## 5. Effect of Binarity on Pulsations

Theoretical (e.g., [27]) and observational (e.g., [28]) studies jointly indicate that most binaries with $P_{orb} < 10$ d are tidally locked and predominantly synchronized and circularized. Theoretically, it is expected that tidal effects should affect the pulsations of the components of binary systems. Reference [29] predicted the existence of tilted pulsations in eclipsing binaries.

Numerous instances of observational evidence for the existence of tidally induced tilted pulsation modes in detached binary stars have recently been reported ([30–33]), including specific references to oEA stars ([34,35]).

In these stars, the pulsation axis is aligned with the tidal axis, which is determined by the presence of the other component. A theoretical analysis of tilted pulsations and their expected orbital modulations was considered by [30]. For example, V1031 Ori, a triple system including a detached eclipsing binary system consisting of two A-type stars, displays tilted dipole pulsations of one component and an aspect-dependent modulation of the pulsation amplitude over the orbital phase with a 180-degree pulsation phase switch in the minima ([33]). Figure 5 shows the TESS light curve of V1031 Ori, exhibiting orbital-phase-dependent modulation of the pulsation amplitude caused by the visible aspect of the pulsation pole and a well-defined, superimposed additional modulation of the pulsation amplitude (a reduction in amplitude) around an orbital phase of 0.5, caused by the so-called "periodic spatial filter (PSF)" effect modeled and discussed previously by [22]. The PSF effect holds substantial potential for in-eclipse pulsation mode identification, which is one of the basic unsolved problems of practical asteroseismology. Another example of the PSF effect, resulting in amplitude magnification in the primary, is shown for the pulsations of RZ Cas in Figure 6. The detailed investigation of hundreds of precise light curves of eclipsing binaries accumulated during the TESS mission will clarify the proportional incidence of tilted pulsations in oEA stars, as well as the spatial structure of the excited modes in these stars.

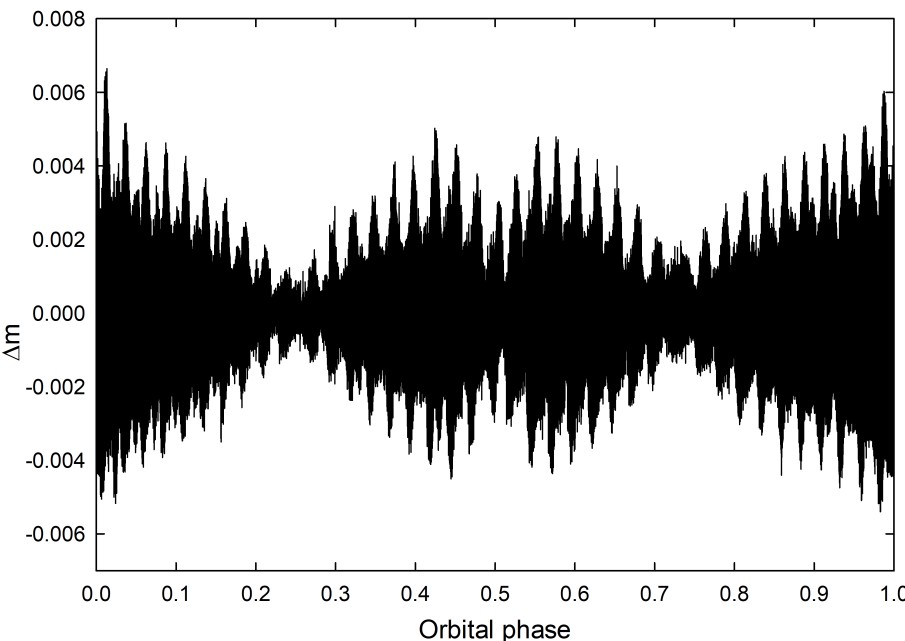

**Figure 5.** Variation of observed pulsation amplitude in V1031 Ori over an orbital period, caused by tilted l = 1, m = 0 nonradial pulsations. The maximum pulsation amplitude is reached at phase 1.0, when the pole of the pulsation axis is directed towards the observer. The expected maximum of the pulsation amplitude at phase 0.5, when the opposite pulsation pole is visible, is actually replaced with a minimum because of the PSF effect hitch is caused by the transit of the primary non-pulsating component in front of the pulsating secondary star.

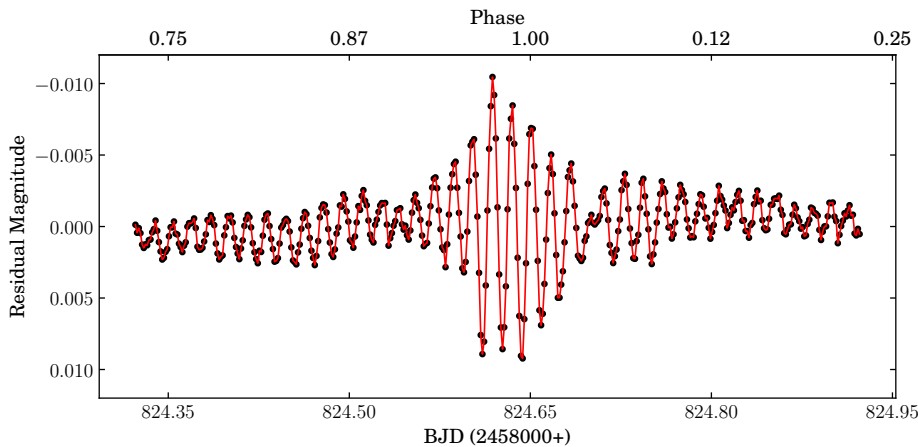

**Figure 6.** A portion of the TESS residual light curve of RZ Cas around the primary minimum at BJD=2458824.6540, displaying the amplitude amplification due to the PSF effect.

## 6. The Active Algol System RZ Cas—A Vital Case Study of the Peculiarities of oEA Stars

As mentioned above, a unique property of oEA pulsators is the co-existence of mass accretion and pulsations, where we expect that the accretion will interact with and modify the characteristics of the pulsations. Since the discovery of the class of oEA stars, only the active and bright (V = 6.24 mag) Algol-type system RZ Cas has been intensively studied both spectroscopically and photometrically (see [22,36–40]). RZ Cas is a bright eclipsing binary system, optimally suited for spectroscopic studies as a representative case elucidating the basic properties of oEA stars. The system consists of an A3V primary with $M_1 = 1.951 \pm 0.005$ $M_\odot$ and a K0 III Roche-lobe-filling secondary component with $M_2 = 0.684 \pm 0.001$ $M_\odot$, separated by a = 6.546 $\pm$ 0.005 $R_\odot$ [40]. The Delta-Scuti-type pulsations in RZ Cas were detected by [41,42]. Reference [36] presented the first spectroscopic evidence for pulsations in oEA stars, using precise radial velocity measurements. Reference [17] reported on abrupt changes in the pulsation spectrum and amplitude in 2001 and suggested that these changes were caused by high mass transfer events. For this reason, this system was monitored from 2001 to 2019 using high-resolution spectroscopy, as well as photometry. In the following subsection, we present the basic results obtained for this system, which we surmise to be representative of the class of oEA stars.

### 6.1. Cyclic Orbital Period Changes in RZ Cas

References [22,40] studied the variations of the orbital period of RZ Cas over a period of two decades and found cyclic $\pm 1$ s variations of the orbital period on a timescale of 6–9 years. The particular cycle covering the interval of 2001–2012 had a duration of approximately 9 years. Spectroscopic observations by [36,37] demonstrated that RZ Cas was in its active state during 2001, with an intensive mass transfer rate, whereas the system was in its quiescent stage during 2006. Two estimations of the mass transfer rate in 2001 by [22,40] produced values of $\dot{M} = 1.3 \times 10^{-6} M_\odot yr^{-1}$ and $1.5 \times 10^{-6}$ $M_\odot yr^{-1}$, respectively.

### 6.2. The 3D Simulation of the Variable Mass Transfer and the Structure of Gas Flows

As already mentioned, Reference [15] included variations in the mass transfer rate in their code for the hydrodynamic simulations of mass transfer from the L1 point, to accommodate the varying location of this point in Kurucz's [16] model atmospheres for the donor. This approach for controlling the mass transfer rate in hydrodynamic simulations corresponds to our interpretation of the dynamics of the L1 spot as a product of the cyclic variability of magnetic activity. We adopted this method for our hydrodynamic simulations of mass transfer in the following oEA stars: RZ Cas, TW Dra, and AS Eri [43]. Using this approach, we can also simulate short-term mass transfer bursts in our code.

For demonstration purposes, we show an example of preliminary 3D simulations of the circumbinary gas flows in RZ Cas with a mass transfer rate of $1.1 \times 10^{-9} M_\odot yr^{-1}$ in Figure 7. The 3D simulations show the variation of the rate of the mass transfer that proceeds via the L1 point of the Roche-lobe-filling star and the density variations in the circulating annulus of gas around the gainer. The variable column density of the circumprimary gas flows projected onto the surface of the gainer can lead to several effects, which are discussed in the sections that follow.

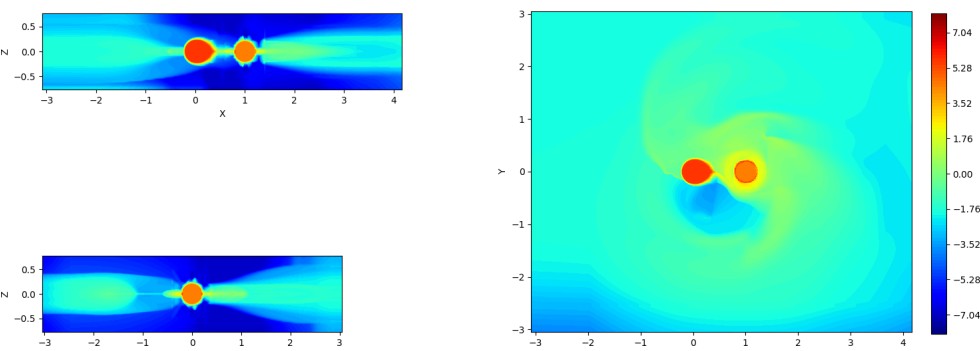

**Figure 7.** The 3D hydrodynamic simulation of the mass transfer in the oEA star RZ Cas in the XYZ coordinate system where the origin of the Z-axis is located at the center of the Roche-lobe-filling star and the X- and Y-axes lie in the orbital plane. The right panel shows the X-Y and the left panel the X-Z and Y-Z cross-sections of the density distribution. The density scale is logarithmic in units of $10^{-11}$ particles per cubic centimeter. The mass transfer generates the circumprimary and circumbinary gas flows forming the quasi-disc in the orbital plane, with an inhomogeneous density that depends on the mass transfer rate.

*6.3. Non-Stationary Mass Transfer and the Variable Asymmetry of the Rossiter–McLaughlin Effect*

The high-resolution spectroscopy of RZ Cas analyzed by [36,37] revealed that RZ Cas experienced a transient phase of rapid mass transfer (RMTP) in 2001 and that an accretion annulus was formed around the primary during the 2000/2001 season, as confirmed by [38]. As noted in [22], the RMTP in 2000/2001 was confirmed independently through an orbital period jump of about +1 s and was interpreted by [22,37] as a high mass transfer event from the secondary to the primary. In 2006, according to the spectroscopic results of [37,40], RZ Cas exhibited a quiescent phase of mass transfer.

The active and quiescent phases in RZ Cas were independently confirmed through their optical effects. Reference [36] observed the asymmetric Rossiter–McLaughlin effect (RME) in RZ Cas occurring during the RMTP in the 2001 season; it was almost symmetric in 2006 during a quiescent stage. The asymmetry of the RME in detached eclipsing binaries and during exoplanet transits is observed when the rotation axis of the star is inclined to the orbital axis of the transiting body. This does not apply to the components of close Algol-type binaries, which are tidally locked and whose rotation axes are perpendicular to the orbital plane.

Reference [37] attributed the asymmetric RME to asymmetric screening and the attenuation of the flux received from the surface of the primary component by an optically thick gas stream and a circulating gas envelope whose effects reached a maximum at the orbital phases preceding the primary minima. Thus, the asymmetric absorption by an optically thick gas stream observed in 2001 led to asymmetry in the shape of the RME, reducing the amplitude of the positive velocity deviation. In 2006, when the mass transfer rate was reduced, the optically thin gas stream did not have a strong effect on the positive amplitude deviation and the RME was almost symmetric. This result established that spectroscopic observations of the variable asymmetry of the RME are a valuable indicator of the mass transfer rate and its temporal variability in Algol systems.

### 6.4. Attenuation of Pulsations Due to the Gas Stream and Equatorial Gas Annulus

The 3D hydrodynamic simulations show that any increase in the rate of the mass transfer changes the dimensions and the density of the gas stream, the gas stream impact zone on the atmosphere, and the circumprimary equatorial gas "belt", which may become optically thick. These effects are expected to be orbital-phase-dependent, with the ability to increase flux extinction across the equatorial zones of the visible surface of the pulsating gainer and, thereby, influence the visibility of nonradial pulsation modes. This effect, as detected by [36,37] in the pulsations of RZ Cas for epochs with high and low mass transfer, was first interpreted by [22]. The attenuation effects of the gas stream and the annulus can also change the shape of the PSF effect and should be accounted for in applications related to pulsation mode identification.

### 6.5. Evidence of the 18-Year Magnetic Cycle—The Detection, Variability, and Migration of Opposite Spots on the Gainer

Using precise high-resolution spectroscopy of RZ Cas, Reference [40] demonstrated that the variations of radial velocities with the orbital phase can be modeled by including two oppositely located cool spots on the surface of the secondary component facing the L1 and L2 points (the L1 and L2 spots). This interpretation provides good agreement with the earlier spectroscopic detection of one spot around L1 found by [38] using spectral line profile modeling and also with the theoretical prediction of [44] for the existence of two opposite spots on the donor. The L1 and L2spots on the donor exhibit opposite variations in their sizes with a period of 9 years, while the L2 spot also exhibits a migration in longitude, returning to its position detected in 2001 after an 18-year cycle. These authors assumed that in the case of RZ Cas, similar structures were observed in the Roche-lobe-filling component as those seen in cool, rapidly rotating RS CVn binaries or single FK-Com- and BY-Dra-type variables. The oppositely located spots in these stars have different intensities and exhibit a switching of activity from one spot to another on time scales of years or decades, known as the flip-flop effect (see [45–48]). Consequently, Reference [40] concluded that the last complete magnetic cycle in the secondary component of RZ Cas lasted 18 years, with 9-year cycles of opposite variation of the magnetic field in the spots.

### 6.6. The Wilson Effect in the L1 Spot as the Possible Mechanism Controlling the Mass-Transfer

Reference [40] studied the interrelations of spot size and mass transfer activity and reported the important discovery of an anti-correlation of the L1 spot size with the mass transfer activity in RZ Cas. They discovered that, for the quiescent years of 2006–2007 and 2015–2016, the L1 spot size was approximately 40 degrees, while in the active year 2001 (and possibly, around 2011, when observations were absent), the spot size was approximately 20 degrees. In order to interpret the mass transfer variations in RZ Cas in terms of an analogy between the structure of sunspots and the physical structure of the RZ Cas donor, these authors proposed a mechanism involving the variable feeding of mass transfer by the L1 spot. It is well known in solar physics that sunspots show a saucer-shaped depression in the photosphere caused by the Lorentz force of the strong magnetic field in the spot, the so-called Wilson depression ([49]). Inside the spot, the level where the optical depth becomes unity ($\tau = 1$) is located well below the level in the photosphere outside the spot. For the Sun, the geometric depth of the depression is of the order of 600–1000 km (e.g., [50–52]). Reference [40] suggested that the existence of such atmospheric depressions close to the L1 point would entail that L1 is "fed" by atmospheric layers of lower density, and therefore, the mass transfer becomes suppressed as the depression becomes deeper. In an earlier section, we discussed how 3D simulations indicate that the variability of the depth of the L1 point in the feeding atmosphere of a donor can lead to a variable mass transfer rate. In reality, the strength of the local magnetic field in the L1 spot controls the depth of the depression and, thus, the density of atmospheric layers that move outward and inward under the variations of the local magnetic field in the spot and can variably feed the L1 point. In this way, the mass transfer rate through L1 is controlled by the magnetic cycle

and the magnetic field at the spot. Data obtained for RZ Cas fit this hypothesis perfectly, displaying the lowest mass transfer activity in 2006–2007 and in 2015–2016, when the spot size and strength were twice as large as in 2001.

### 6.7. The Time Variability of Pulsation Spectra

The abrupt variability of both the photometric pulsation spectrum and the pulsation amplitude of the dominant mode of RZ Cas was reported in 2001 by [17], who proposed that these changes could be caused by episodes of high mass transfer/accretion. Monitoring of the RZ Cas system during the following decade demonstrated that the amplitude of pulsation of the dominant mode was recovered to approximately the previous values in 2004–2006 and, then, declined again to a value of approximately 1 mmag in 2009. The complete analysis of the accretion-driven variation of the dominant pulsation mode was discussed by [22] and by [40], who demonstrated that the 9-year-long cycle of pulsation amplitude variability is related to the variable mass transfer rate caused by the 18-year-long full cycle of the magnetic activity of the donor star.

Long-term space-based observations of dozens of oEA stars will permit us to monitor their pulsation amplitude variations as a function of time and deepen our understanding of the role of magnetic cycles and mass transfer events in driving the variability of the physical and pulsational properties of mass-accreting stars.

## 7. Discussion

Observations in recent years have confirmed that it is highly probable that accretion affects the pulsations of oEA stars, as predicted at the time of discovery of this class of stars. Cyclical variations in the amplitudes of pulsations and in pulsation spectra themselves have been discovered. This cyclical behavior correlates with the observed manifestations of cyclical behavior in magnetic activity, in particular the behavior of spot activity in the L1 region. The proposed mechanism for variations in the mass transfer rate in the form of the Wilson effect, which was previously used in hydrodynamic simulations, confirms the feasibility of this hypothesis in actual binary systems. Variations in the rates of matter transfer create a circulation in the equatorial zone of the gainer, in the form of an optical disk or a quasi-disk with a variable optical density, which optically affects the observed pulsation amplitudes, as well as the asymmetry of the RME. Eclipses in the system provide a unique opportunity for identifying pulsation modes. The aforementioned characteristics of pulsations in oEA stars, coupled with the methods of asteroseismology, provide new diagnostic possibilities for interpreting the variability of the rates of matter transfer and, consequently, magnetic activity in binary systems. The doubling of the number of known pulsating oEA stars by analyzing high-precision data from space telescopes makes it possible to study these effects on a large sample of systems with diverse physical and pulsational characteristics.

**Author Contributions:** Conceptualization, D.M.; methodology, D.M., K.G., H.L., and A.T.; software, K.G., H.L., V.N., and A.T.; formal analysis, D.M. and H.L.; investigation, D.M. and K.G.; data curation, D. M., K.G., H.L., and C.E.; writing—original draft preparation, D.M. and C.E.; writing—review and editing, D.M. and C. E.; visualization, D.M.; supervision, D.M.; project administration, D.M. All authors have read and agreed to the published version of the manuscript

**Funding:** This research received no external funding.

**Institutional Review Board Statement:** Not applicable.

**Informed Consent Statement:** Not applicable.

**Acknowledgments:** D.E.M. acknowledges his work as part of the research activity supported by the National Astronomical Research Institute of Thailand (NARIT). The research leading to these results has (partially) received funding from the KU Leuven Research Council (Grant C16/18/005: PARADISE) and from the BELgian federal Science Policy Office (BELSPO) through PRODEX Grant PLATO. The SALT Time Allocation Committee is thanked for ample observing time.

**Conflicts of Interest:** The authors declare no conflict of interest.

## Abbreviations

The following abbreviations are used in this manuscript:

| | |
|---|---|
| 3D | Three-dimensional |
| c/d | Cycles per day |
| DFT | Discrete Fourier transform |
| EA/D | Eclipsing Algol detached configuration |
| EA/SD | Eclipsing Algol semi-detached configuration |
| EA/DM | Eclipsing Algol main sequence components |
| EB | Eclipsing binaries with Beta Lyrae-type light curve |
| EA | Eclipsing binaries with Algol-type light curve |
| NRP | Nonradial pulsations |
| L1 | First Lagrange point |
| MS | Main sequence |
| m | Metallic line spectrum in Sp |
| mag | Magnitude |
| oEA | Oscillating eclipsing Algols |
| PSF | Periodic spatial filter |
| RMTP | Rapid mass transfer phase |
| RME | Rossiter–McLaughlin effect |
| SALT | South African Large Telescope |
| Sp | Spectral classification |
| TESS | Transiting Exoplanet Survey Satellite |

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
