# Peer review of "Mass-Accreting Pulsating Components of Algols"

_galaxies, doi:10.3390/galaxies10050097_

Round 1
Reviewer 1 Report
This paper is a qualitative review on the influence of mass transfer on the pulsations of oEA stars. The paper is well written, the systems selected as examples are fine, while the figures are well presented and informative. However, there are a few things that need to be clarified before it is accepted.
My comments are given below (Com.=comment, P=page, ln=line) and although they are numerous, their majority concern typos and syntax errors, except comments 40-48, which should be handled carefully.
Comments
Com. 1 (p1, l5): “85 new pulsating components”: I found out that significant number of systems (29 up to 38 out of 85) in Table 1 are previously known cases. Please check carefully the comment No 40 and use the correct number of new discoveries or rephrase the sentence in the abstract. Moreover, this sentence comes in a slight contradiction with the first sentence of the abstract that mentions that this paper concern the Semi-detached systems, while in Table 1 many detached ones appear also. Please rephrase.
Com. 2 (p1, ln13): “diagram” à Change to “diagram (e.g. \delta Sct, \gamma Dor etc.)”.
Com. 3 (p1, ln15): “The pulsating” à Change to “Many pulsating”.
Com. 4 (p1, ln15): “Lambda” à Use the greek letter like you did for the gamma Doradus and delta scuti.
Com. 5 (p1, ln18): “augmenting the” à Change to “augmenting also the”.
Com. 6 (p1, ln29): “orbits” à Change to “orbital elements”.
Com. 7 (p1, ln32): “- an” à Change to “- one”.
Com. 8 (p2, ln33): “than obtained” à Change to “than that obtained”.
Com. 9 (p2, ln36): “either or” à Change to “either one or”.
Com. 10 (p2, ln38-39): “and the application… stellar interiors” à The meaning is not clear. Please rephrase add better split the whole sentence into two parts since it is too long (ln 36-39).
Com. 11 (Figure 1): The roche lobe of the primary is hardly seen in a printed version. I recommended to use red dotted/dashed lines for the roche lobes boundaries of both components if possible.
Com. 12 (p2, ln41-42): “of spectral….hereafter)” à Interchange as “B, A, or F (the primary, hereafter) spectral type”.
Com. 13 (p2, ln48): “class be called the” à Change to “class should be called as”.
Com. 14 (p2, ln49): “these stars” à Change to “them”.
Com. 15 (p2, ln52-53): “Algol system with an oEA component”à I believe that there is a bit of redundancy of the Algol term here. The term “oEA” includes the term Algol. I recommend to change to “A particular oEA star with a \delta Sct component is depicted in….”.
Com. 16 (p2, ln55-56): “and in a … oEA stars”à This is not clear. This sentence is bonded to the sentence “We present…identified”. Which selection of well-studied oEA stars do you mean? Maybe the discussion of RZ Cas? If this is the case, please change to “and in a representative well-studied case (RZ Cas)”.
Com. 17 (p2, ln61-62): “correspond…transfer.”à Change to “correspond to seasonal variations of the mass transfer rate”.
Com. 18 (p2, ln63): “and the importance of oEA stars for studies”à Change to “their importance for studies”.
Com. 19 (p3, ln68): “by the”à delete “the”.
Com. 20 (p3, ln70): “originally”à change to “initially”.
Com. 21 (p3, ln73): “interior structure”à change to “internal structure” or keep only the “interior” (delete “structure”).
Com. 22 (p3, ln75): “diagrams”à change to “diagram”.
Com. 23 (p3, ln76): Define or expand “MS”. Add it in the list of abbreviations.
Com. 24 (p3, ln80): “found”à change to “revealed”.
Com. 25 (p3, ln88): “in the … which”à change to “in detached systems, which”.
Com. 26 (p3, ln93): “other A-F”à change to “other single A-F”.
Com. 27 (p3, ln99-100): The cyclic changes in orbital period can be also explained by the existence of tertiary component. Please add this also for completeness’ reasons.
Com. 28 (p3, ln105): “in distance”à Change to “in the distance”.
Com. 29 (p3, ln108-112): This sentence is too long. Please split it into two or three shorter ones.
Com. 30 (p3, ln113): “of variations…rate”à Interchange as “mass-transfer rate variations”.
Com. 31 (p3, ln115): Delete “discussed in” and include the reference in a parenthesis ([14]).
Com. 32 (p3, ln115): “by [15] who used”à Split the sentence “by [15]. In the latter study, the authors…”.
Com. 33 (p3, ln117-120): “Their calculations…point”à This is a long sentence and difficult to understand. Please split it or rephrase.
Com. 34 (p4, ln128): “to very”à change to “with very”
Com. 35 (p4, ln153):”controlling” à I believe that “controlling” is incorrect. I recommend “triggering” or “leading” or “describing”.
Com. 36 (p4, ln157-158): “selected active Algols” à The word “active” is not clear. Do you mean magnetically active or with active mass transfer? Please clarify.
Com. 37 (p5, ln181-183): “The majority… 10 mmag”à This result needs proper references or a relevant diagram that shows what the authors claim.
Com. 38 (p5, ln185): delete “oEA stars discovered”.
Com. 39 (p5, ln200 and Fig. 2): “(the amplitudes…eye)”à Please delete this and add either an internal panel in Fig. 2 showing the pulsations or add a new figure. The sentence in line 200 describes something that is not shown. Moreover, change the type of the BJD value from exponential to simple numerical (i.e. 2458700+).
Com. 40 (table 1): As already mentioned in comment #1, there are a lot of cases given in Table 1 that are not new discoveries, but they are previously known cases. These systems should be given in another table with different header that will not include the words “discovered by us”. The new table may have a header like “re-analysed cases” or something similar. Particularly, I refer to the following systems (in papers from 2020 and later, the systems might be given with their TESS name):
CPD-30 740, GP Cet, SW PsA, KP Eri, AU For, V483 Vel, BH Scl, AK Eri, RU Gru, TW Pup, V2541 Cyg, AB Vol, KZ Eri, HD 69683, AW Men, CI CVn, V342 And : Xiang-dong Shi et al., 2022, ApJS, 259, 50
CPD-30 740 and TYC 683-640-1: B. Ulas et al., 2022, Ap&SS, 367, 22
BR Hyi: G. Pojmanski, 2002, AcA, 52, 397; Sharma, S., et al., 2018, MNRAS, 473 ,2004
X Tri: G. Turner et al., 2014, JAAVSO, 42, 134
AW Vel: D.J.W. Moriarty et al., 2013, JAAVSO, 41, 182; Liakos&Niarchos, 2017, MNRAS, 465, 1181
RZ Mic: M. Streamer et al., 2016, JAAVSO, 44, 39; Liakos&Niarchos, 2017, MNRAS, 465, 1181
IO UMa: F. Soydugan et al., 2013, MNRAS, 432, 3278
UW Cyg: A. Liakos et al., 2012, MNRAS, 422, 1250
X Pic : C. von Essen et al., 2020, AJ, 160, 34
V389 Cas: D. Korda et al., 2015, NewA, 40, 64
HD 34954: J. W. Lee et al., 2019, AJ, 157, 223
V961 Cep: I. Volkov, 2022, PZ, 1, 1
TT Hor: D.J.W. Moriarty et al., 2013, JAAVSO, 41, 182; M. Streamer et al., 2018, MNRAS, 480, 1372
BB Cep, NO Com, V359 Her, V1050 Her, KL Per, V735 Cep, GT Cas, AL Cam: F. Kahraman-Alicavus et al., 2022arXiv220412952K (submitted to Research in Astronomy and Astrophysics)
For the last set of systems by Kahraman-Alicavus et al., I do not insist to exclude them from the “discoveries” table, since that paper appears only as submitted and not officially accepted or published yet. However, this should be re-check by the date of the publication of the present paper.
Com. 41 (table 1): It is not clear how the authors classify the light curves types (EA/EB), the roche geometry (D, SD), and the spectral types of the systems. These should be addressed in the text or proper references should be given.
Com. 42 (table 1): In many systems there is the designator “S”. This should be added in the abbreviations list, since, as I understand, this might be different from the “SD” (semi-detached) or the authors mistyped “SD”. The latter (SD) is wrongly given in the abbreviations list.
Com. 43 (table 1): Please use the lower case “f” for the frequency in the last column, since the capital “F” is usually used to denote quantities like force. Moreover, please use “c/d” since it is already defined in the list of abbreviations.
Com. 44 (table 1): Please add errors in the orbital periods and frequencies. Alternatively, you can mention in the header of the table or in the text how large are these errors. I assume that you have a 5-digit precision for the orbital periods and a 2-digit precision for the pulsation frequencies.
Com. 45 (table 1): For HD29766 a type “Star” is given. This should be corrected or explained in the text or as a tablenote.
Com. 46 (table 1): In many systems there is the letter “m” in their spectral types. This should be explained in the text or as a tablenote.
Com. 47 (table 1): Add a gap between “KZ” and “Eri” in the KZ Eri’s name.
Com. 48 (table 1): For W Vol, CI CVn, ES Ori, TT Hor, and V343 Lac delete the gap between the spectral and luminosity classes (e.g. A0 V à A0V).
Com. 49 (Figure 5): add a gap between “=1,” and “m=0” in the legend.
Com. 50 (Figure 5): If I correctly understand, the pulsating component in V1031 Ori is the secondary (i.e. the one that is eclipsed during phase 0.5). If this is the case please add this explanation in the legend or if it is not, please clarify.
Com. 51 (Figure 6): It would be nice to add the orbital phase values in the upper x-axis.
Com. 52 (p10, ln253): add “mag” after “V=6.24”.
Com. 53 (p10, ln255): “study”à change to “studies”.
Com. 54 (p10, ln258): delete the gap between “=” and “0.684”.
Com. 55 (p10, ln261): delete the gap before “[17]”.
Com. 56 (p10, ln263): delete the gap before “was”.
Com. 57 (p11, ln299): “spectroscopy presented by”à change to “the spectroscopic results of”.
Com. 58 (p11, ln319): “by the”à change to “due to”.
Com. 59 (p12, ln326-327): “effect, as detected…and [36]”à change to “effect, as detected by [35] and [36] in the pulsations of RZ Cas for epochs with high and low mass transfer, was first…”.
Com. 60 (p12, ln333): Expand “RV” or add it in the table of abbreviations.
Com. 61 (p12, ln337): “by [43] of” à change to “of [43] for”.
Com. 62 (p12, ln338): The term “opposite spots” is not clear. Do you mean hot spots on the primary? Please, clarify.
Com. 63 (p12, ln347): The term “opposite activity” is not clear. Please, clarify.
Com. 64 (p12, ln361): “depth t=1” à change to “depth becomes unity (t=1)” (use the symbol \tau as you did).
Com. 65 (p12, ln367): “an L1” à change to “the L1”.
Com. 66 (p13, ln412-418): Check again carefully the funding statements. In ln412 says that D.M. and C.E (by the way delete the gap between “C.” and “E.”) acquired funding. In ln414 says that no funding received and in ln417-418 says that funding was acquired. Please check again in order to avoid any misunderstandings.
Com. 67 (p14, list of abbreviations): Check my previous comments in order to include more abbreviated terms used in the text. There is something wrong with the justification and tabulation from EA/D up to EB lines. NRP abbreviation should be move in the next line.
Author Response
Reply to referee 2.
We thank referee 1 for his detailed work and comments on the manuscript. All of his comments were accepted and listed in our reply.
The only comment about changing the referencing style (after consultations with the journal Editor) was neglected.
We did a large number of text editions, the longest modified sentences we present in the text by red font color.
Referee:
Comments and Suggestions for Authors
This paper is a qualitative review of the influence of mass transfer on the pulsations of oEA stars. The paper is well written, the systems selected as examples are fine, and the figures are well presented and informative. However, there are a few things that need to be clarified before it is accepted.
My comments are given below (Com.=comment, P=page, ln=line) and although they are numerous, their majority concern typos and syntax errors, except comments 40-48, which should be handled carefully.
Comments
Referee:
Com. 1 (p1, l5): “85 new pulsating components”: I found out that a significant number of systems (29 up to 38 out of 85) in Table 1 are previously known cases. Please check carefully comment No 40 and use the correct number of new discoveries or rephrase the sentence in the abstract. Moreover, this sentence comes in a slight contradiction with the first sentence of the abstract that mentions that this paper concern the Semi-detached systems, while in Table 1 many detached ones appear also. Please rephrase.
Authors: Done
Referee:
Com. 2 (p1, ln13): “diagram” à Change to “diagram (e.g. \delta Sct, \gamma Dor etc.)”.
Authors: Done
Referee:
Com. 3 (p1, ln15): “The pulsating” à Change to “Many pulsating”.
Authors: Done
Referee:
Com. 4 (p1, ln15): “Lambda” à Use the greek letter like you did for the gamma Doradus and delta scuti.
Authors: Done
Referee:
Com. 5 (p1, ln18): “augmenting the” à Change to “augmenting also the”.
Authors: Done
Referee:
Com. 6 (p1, ln29): “orbits” à Change to “orbital elements”.
Authors: Done
Referee:
Com. 7 (p1, ln32): “- an” à Change to “- one”.
Authors: Done
Referee:
Com. 8 (p2, ln33): “than obtained” à Change to “than that obtained”.
Authors: Done
Referee:
Com. 9 (p2, ln36): “either or” à Change to “either one or”.
Authors: Done
Referee:
Com. 10 (p2, ln38-39): “and the application… stellar interiors” à The meaning is not clear. Please rephrase add better split the whole sentence into two parts since it is too long (ln 36-39).
Authors: We shorten this sentence as follows: "Furthermore, when pulsations of either one or both components of eclipsing binary systems are detected, the accurately-determined physical parameters can be used very effectively for the refinement of the pulsation model."
Referee:
Com. 11 (Figure 1): The roche lobe of the primary is hardly seen in a printed version. I recommended to use red dotted/dashed lines for the roche lobes boundaries of both components if possible.
Authors: We replaced Figure 1 with a new one having brighter Roche lobe boundaries.
Referee:
Com. 12 (p2, ln41-42): “of spectral….hereafter)” à Interchange as “B, A, or F (the primary, hereafter) spectral type”.
Authors: Done
Referee:
Com. 13 (p2, ln48): “class be called the” à Change to “class should be called as”.
Authors: Done
Referee:
Com. 14 (p2, ln49): “these stars” à Change to “them”.
Authors: Done
Referee:
Com. 15 (p2, ln52-53): “Algol system with an oEA component”à I believe that there is a bit of redundancy of the Algol term here. The term “oEA” includes the term Algol. I recommend to change to “A particular oEA star with a \delta Sct component is depicted in….”.
Authors: We replaced this sentence as follows: " A particular oEA star with the pulsating component is depicted in Figure 1".
Referee:
Com. 16 (p2, ln55-56): “and in a … oEA stars”à This is not clear. This sentence is bonded to the sentence “We present…identified”. Which selection of well-studied oEA stars do you mean? Maybe the discussion of RZ Cas? If this is the case, please change to “and in a representative well-studied case (RZ Cas)”.
Authors: We replaced this sentence as follows: "We present a concise review of the most recent observational data and analytical results associated with pulsations identified in eclipsing binaries observed by TESS and in representative well-studied case (RZ Cas).
Referee:
Com. 17 (p2, ln61-62): “correspond…transfer.”à Change to “correspond to seasonal variations of the mass transfer rate”.
Authors: Done
Referee:
Com. 18 (p2, ln63): “and the importance of oEA stars for studies”à Change to “their importance for studies”.
Authors: Done
Referee:
Com. 19 (p3, ln68): “by the”à delete “the”.
Authors: Done
Referee:
Com. 20 (p3, ln70): “originally”à change to “initially”.
Authors: Done
Referee:
Com. 21 (p3, ln73): “interior structure”à change to “internal structure” or keep only the “interior” (delete “structure”).
Authors: Done
Referee:
Com. 22 (p3, ln75): “diagrams”à change to “diagram”.
Authors: Done
Referee:
Com. 23 (p3, ln76): Define or expand “MS”. Add it in the list of abbreviations.
Authors: Done
Referee:
Com. 24 (p3, ln80): “found”à change to “revealed”.
Authors: Done
Referee:
Com. 25 (p3, ln88): “in the … which”à change to “in detached systems, which”.
Authors: Done
Referee:
Com. 26 (p3, ln93): “other A-F”à change to “other single A-F”.
Authors: Done
Referee:
Com. 27 (p3, ln99-100): The cyclic changes in orbital period can be also explained by the existence of tertiary component. Please add this also for completeness’ reasons.
Authors: We add to the sentence " or by the existence of tertiary component"
Referee:
Com. 28 (p3, ln105): “in distance”à Change to “in the distance”.
Authors: Done
Referee:
Com. 29 (p3, ln108-112): This sentence is too long. Please split it into two or three shorter ones.
Authors: Done
Referee:
Com. 30 (p3, ln113): “of variations…rate”à Interchange as “mass-transfer rate variations”.
Authors: Done
Referee:
Com. 31 (p3, ln115): Delete “discussed in” and include the reference in a parenthesis ([14]).
Authors: Done
Referee:
Com. 32 (p3, ln115): “by [15] who used”à Split the sentence “by [15]. In the latter study, the authors…”.
Authors: Done
Referee:
Com. 33 (p3, ln117-120): “Their calculations…point”à This is a long sentence and difficult to understand. Please split it or rephrase.
Authors: We re-phrase this sentence as follows: "Their calculations demonstrated that the degree of Roche-lobe overflow and the rate of mass transfer observed in Algols are dependent on the height of the location of the inner Lagrangian point in the atmospheric layers of the donor."
Referee:
Com. 34 (p4, ln128): “to very”à change to “with very”
Authors: Done
Referee:
Com. 35 (p4, ln153):”controlling” à I believe that “controlling” is incorrect. I recommend “triggering” or “leading” or “describing”.
Authors: Done
Referee:
Com. 36 (p4, ln157-158): “selected active Algols” à The word “active” is not clear. Do you mean magnetically active or with active mass transfer? Please clarify.
Authors: Done as follows: "Consequently, there is a need for detailed observations of selected magnetically active Algols and for novel observational indicators of magnetic cycles and strong
mass-transfer events."
Referee:
Com. 37 (p5, ln181-183): “The majority… 10 mmag”à This result needs proper references or a relevant diagram that shows what the authors claim.
Authors: The diagram with the distribution of pulsation amplitudes of oEA stars is a mess and doesn't reflect well the physics of pulsations. The real surface pulsation amplitudes can be well different from observed disk-integrated values and depend on modal spatial structure, surface amplitude cancellation effect, the inclination of the rotation, and the pulsation axes. The typically observed amplitudes of pulsations of Delta Scuti-type stars are well known and are less than 0.05 mag with a rare exception for evolved high-amplitude pulsators (not the case of oEA stars). Instead of showing a diagram, we give a reference to two papers Liakos & Niarchos 2017 (EA/DSCT and oEA) and Mkrtichian et al., 2018 (only oEA) where the reader can find Tables with typical observed amplitudes and periods.
Referee:
Com. 38 (p5, ln185): delete “oEA stars discovered”.
Authors: Done
Referee:
Com. 39 (p5, ln200 and Fig. 2): “(the amplitudes…eye)” Please delete this and add either an internal panel in Fig. 2 showing the pulsations or add a new figure. The sentence in line 200 describes something that is not shown. Moreover, change the type of the BJD value from exponential to simple numerical (i.e. 2458700+).
Authors: Done as follows " Figure 2 shows the TESS binary light curve of RR Dra containing very low-
amplitude oscillations hidden in the observational scatter. "
Referee:
Com. 40 (table 1): As already mentioned in comment #1, there are a lot of cases given in Table 1 that are not new discoveries, but they are previously known cases. These systems should be given in another table with different header that will not include the words “discovered by us”. The new table may have a header like “re-analysed cases” or something similar. Particularly, I refer to the following systems (in papers from 2020 and later, the systems might be given with their TESS name):
CPD-30 740, GP Cet, SW PsA, KP Eri, AU For, V483 Vel, BH Scl, AK Eri, RU Gru, TW Pup, V2541 Cyg, AB Vol, KZ Eri, HD 69683, AW Men, CI CVn, V342 And : Xiang-dong Shi et al., 2022, ApJS, 259, 50
CPD-30 740 and TYC 683-640-1: B. Ulas et al., 2022, Ap&SS, 367, 22
BR Hyi: G. Pojmanski, 2002, AcA, 52, 397; Sharma, S., et al., 2018, MNRAS, 473 ,2004
X Tri: G. Turner et al., 2014, JAAVSO, 42, 134
AW Vel: D.J.W. Moriarty et al., 2013, JAAVSO, 41, 182; Liakos&Niarchos, 2017, MNRAS, 465, 1181
RZ Mic: M. Streamer et al., 2016, JAAVSO, 44, 39; Liakos&Niarchos, 2017, MNRAS, 465, 1181
IO UMa: F. Soydugan et al., 2013, MNRAS, 432, 3278
UW Cyg: A. Liakos et al., 2012, MNRAS, 422, 1250
X Pic : C. von Essen et al., 2020, AJ, 160, 34
V389 Cas: D. Korda et al., 2015, NewA, 40, 64
HD 34954: J. W. Lee et al., 2019, AJ, 157, 223
V961 Cep: I. Volkov, 2022, PZ, 1, 1
TT Hor: D.J.W. Moriarty et al., 2013, JAAVSO, 41, 182; M. Streamer et al., 2018, MNRAS, 480, 1372
BB Cep, NO Com, V359 Her, V1050 Her, KL Per, V735 Cep, GT Cas, AL Cam: F. Kahraman-Alicavus et al., 2022arXiv220412952K (submitted to Research in Astronomy and Astrophysics)
For the last set of systems by Kahraman-Alicavus et al., I do not insist to exclude them from the “discoveries” table, since that paper appears only as submitted and not officially accepted or published yet. However, this should be re-check by the date of the publication of the present paper.
Authors:
Reference to von Essen about discovery pulsations in X Pic isn't correct, they didn't.
We removed from our Table1 all stars marked by the referee which were recently found as pulsating, now Table 1 includes the list of 49 new pulsators discovered
solely by us (for AI Dra earlier report about the discovery of pulsations by Narusava et al. 2002, was unsafe, we found safely pulsations and accurately measured dominant pulsation frequency). We wrote: "[ 24] listed 57 new pulsating components in EA-type binary systems by analyzing TESS data. In Table 1 we present a list of 49 new oEA and some detached pulsating eclipsing binary stars discovered by us after 2018, based on TESS light curves."
Referee:
Com. 41 (table 1): It is not clear how the authors classify the light curves types (EA/EB), the roche geometry (D, SD), and the spectral types of the systems. These should be addressed in the text or proper references should be given.
Authors: This is a standard classification of eclipsing binaries in GCVS. Done, we rename classification and add to the list of abbreviations
Referee:
Com. 42 (table 1): In many systems there is the designator “S”. This should be added in the abbreviations list, since, as I understand, this might be different from the “SD” (semi-detached) or the authors mistyped “SD”. The latter (SD) is wrongly given in the abbreviations list.
Authors: We renamed classification S to commonly used SD.
Referee:
Com. 43 (table 1): Please use the lower case “f” for the frequency in the last column, since the capital “F” is usually used to denote quantities like force. Moreover, please use “c/d” since it is already defined in the list of abbreviations.
Authors: Done
Referee:
Com. 44 (table 1): Please add errors in the orbital periods and frequencies. Alternatively, you can mention in the header of the table or in the text how large are these errors. I assume that you have a 5-digit precision for the orbital periods and a 2-digit precision for the pulsation frequencies.
Authors: We wrote an explanation about precision in the Table1 caption
Referee:
Com. 45 (table 1): For HD29766 a type “Star” is given. This should be corrected or explained in the text or as a tablenote.
Authors: Done (it was a misprint)
Referee:
Com. 46 (table 1): In many systems there is the letter “m” in their spectral types. This should be explained in the text or as a tablenote.
Authors: This is a commonly used classification for a metallic line spectrum. Explanation about metallic line classification is included in abbreviations.
Referee:
Com. 47 (tabl`e 1): Add a gap between “KZ” and “Eri” in the KZ Eri’s name.
Authors: Done
Referee:
Com. 48 (table 1): For W Vol, CI CVn, ES Ori, TT Hor, and V343 Lac delete the gap between the spectral and luminosity classes (e.g. A0 V à A0V).
Authors: Done
Referee:
Com. 49 (Figure 5): add a gap between “=1,” and “m=0” in the legend.
Authors: Done
Referee:
Com. 50 (Figure 5): If I correctly understand, the pulsating component in V1031 Ori is the secondary (i.e. the one that is eclipsed during phase 0.5). If this is the case please add this explanation in the legend or if it is not, please clarify.
Authors: Yes, the pulsating component is secondary. The caption to figure 5 is now as follows:
"Variation of observed pulsation amplitude in V1031 Ori over an orbital period,
caused by tilted l=1, m=0 nonradial pulsations. The maximum pulsation amplitude is
reached at phase 1.0, when the pole of the pulsation axis is directed toward the observer.
The expected maximum of the pulsation amplitude at phase 0.5, when the opposite
pulsation pole is visible, is actually replaced with a minimum because of the PSF effect
hitch is caused by the transit of the primary non-pulsating component in front of the
pulsating secondary star"
Referee:
Com. 51 (Figure 6): It would be nice to add the orbital phase values in the upper x-axis.
Authors: Done. We replaced Figure 6 with a new one. We added to the caption *residual* to make clearer what kind of curve we show, it looks now as follows:
"A portion of the TESS *residual* light curve of RZ Cas around the primary minimum at BJD=2458824.6540, displaying the amplitude amplification due to the PSF effect.*
Referee:
Com. 52 (p10, ln253): add “mag” after “V=6.24”.
Authors: Done
Referee:
Com. 53 (p10, ln255): “study”à change to “studies”.
Authors: Done
Referee:
Com. 54 (p10, ln258): delete the gap between “=” and “0.684”.
Authors: Done
Referee:
Com. 55 (p10, ln261): delete the gap before “[17]”.
Authors: Done
Referee:
Com. 56 (p10, ln263): delete the gap before “was”.
Authors: Done
Referee:
Com. 57 (p11, ln299): “spectroscopy presented by”à change to “the spectroscopic results of”.
Authors: Done
Referee:
Com. 58 (p11, ln319): “by the”à change to “due to”.
Authors: Done
Referee:
Com. 59 (p12, ln326-327): “effect, as detected…and [36]”à change to “effect, as detected by [35] and [36] in the pulsations of RZ Cas for epochs with high and low mass transfer, was first…”.
Authors: Done
Referee:
Com. 60 (p12, ln333): Expand “RV” or add it in the table of abbreviations.
Authors: Done
Referee:
Com. 61 (p12, ln337): “by [43] of” à change to “of [43] for”.
Authors: Done
Referee:
Com. 62 (p12, ln338): The term “opposite spots” is not clear. Do you mean hot spots on the primary? Please, clarify.
Authors: Done
Referee:
Com. 63 (p12, ln347): The term “opposite activity” is not clear. Please, clarify.
Authors: I replace it as follows ".. exhibit opposite variations in their sizes with a period of 9 years.."
Referee:
Com. 64 (p12, ln361): “depth t=1” à change to “depth becomes unity (t=1)” (use the symbol \tau as you did).
Authors: Done
Referee:
Com. 65 (p12, ln367): “an L1” à change to “the L1”.
Authors: Done
Referee:
Com. 66 (p13, ln412-418): Check again carefully the funding statements. In ln412 says that D.M. and C.E (by the way delete the gap between “C.” and “E.”) acquired funding. In ln414 says that no funding received and in ln417-418 says that funding was acquired. Please check again in order to avoid any misunderstandings.
Authors: Done
Referee:
Com. 67 (p14, list of abbreviations): Check my previous comments in order to include more abbreviated terms used in the text. There is something wrong with the justification and tabulation from EA/D up to EB lines. NRP abbreviation should be moved to the next line.
Authors: Done.
Reviewer 2 Report
The author reviews a small but interesting group of Algols. It is ok that the author uses one well-studied system RZ Cas to demonstrate the characteristics of oEA stars since the necessity of high-resolution spectrometry and space-based photometry makes the observation difficult and the sample very small. But it is still expected to see more statistics information as a review paper. It is not clear from the paper how many of the oEA stars are well studied and how much the progress is. I suggest the author add a paragraph describing the current observational progress or difficulties in oEA study, and how much/what kind of observation is needed to establish a statistically meaningful sample.
some minor issues and suggestions:
1. It is better to write most of(if not all) the citations in the form of (author, year) rather than citation number.
2. Please list the journal name explicitly in the references.
3. page 3 line 117, " their calculation demonstrate...... locate at inner Lagrangian point", the sentence is confusing. The common understanding is the RLOF is through the L1 point, is there any difference here? please be more specific.
4. Fig2 & Fig 5, it is better to plot the original light curve together with the pulsational light curve with the orbital variation removed.
5. Table 1 lists 85 new candidates. It is better to give more statistical information on those objects or the 70 objects listed in paper[23], since it is a review paper rather than a discovery paper.
6.page 6, Figure4, LSD profile should be defined or referenced
7. page10, line 270, 2001-2002, is it a typo for 2001-2020?
Author Response
Reply to referee 2
The referee:
Comments and Suggestions for Authors
The author reviews a small but interesting group of Algols. It is ok that the author uses one well-studied system RZ Cas to demonstrate the characteristics of oEA stars since the necessity of high-resolution spectrometry and space-based photometry makes the observation difficult and the sample very small. But it is still expected to see more statistics information as a review paper. It is not clear from the paper how many of the oEA stars are well studied and how much the progress is. I suggest the author add a paragraph describing the current observational progress or difficulties in oEA study, and how much/what kind of observation is needed to establish a statistically meaningful sample.
some minor issues and suggestions:
1. It is better to write most of(if not all) the citations in the form of (author, year) rather than citation number.
Authors: According to the MDPI standards and recommendations of editors I followed the used form of citations.
2. Please list the journal name explicitly in the references.
Authors: Checked.
3. page 3 line 117, " their calculation demonstrate...... locate at inner Lagrangian point", the sentence is confusing. The common understanding is the RLOF is through the L1 point, is there any difference here? please be more specific.
Authors: We re-phrase this sentence.
Referee:
4. Fig2 & Fig 5, it is better to plot the original light curve together with the pulsational light curve with the orbital variation removed.
Authors:
For these two stars the pulsation amplitude is small compared to a binary one, by these reasons and for better representation of pulsations, we decided to show in Fig.2 and 5 only the residual light curve. A new Figure 2 for RR Dra shows binary and zoomed pulsational curves hidden in the photometric errors. We keep for Fig. 5 only residual curves.
5. Table 1 lists 85 new candidates. It is better to give more statistical information on those objects or the 70 objects listed in the paper[23], since it is a review paper rather than a discovery paper.
Authors: Our plan was to give only short information about new discovered pulsators in this review paper as these systems are not well studied.
A more detailed statistical investigation of the sample will be done later when we collect more information (in the best case - absolute parameters) about these systems. Indeed, according to the recommendations of the
second referee, we did improvements to this list and removed the systems which were actually reported as pulsators in recent papers. Now the list of stars discovered and investigated by us consists of 49 systems.
6.page 6, Figure4, LSD profile should be defined or referenced.
Done: We put the reference to our paper in the press also cited in astro-ph archive:2208.03072v1 about LSD code
7. page10, line 270, 2001-2002, is it a typo for 2001-2020?
Authors: Done, this is a misprint, should be 2001-2012